# Spatiotemporal Evolution of Cultivated Land Use Eco-Efficiency and Its Dynamic Relationship with Landscape Pattern Change from the Perspective of Carbon Effect: A Case Study of Henan, China

Qi Liu 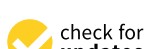, Jiajun Qiao *, Dong Han, Mengjuan Li and Liangxiao Shi

College of Geography and Environmental Science, Henan University, Kaifeng 475004, China;
liuqi@henu.edu.cn (Q.L.)
* Correspondence: jjqiao@henu.edu.cn; Tel.: +86-0371-23881860

**Abstract:** Cultivated land is a vital factor in agricultural production but faces multiple challenges, including declining total area, spatial transformation, and ecological degradation. It is imperative to enhance cultivated land use eco-efficiency (CLUE). This study aimed to evaluate the CLUE that considers both carbon sequestration and emissions using the SBM model at the county level. Next, spatial autocorrelation was employed to measure CLUE's spatial correlation. The spatial agglomeration pattern of CLUE was determined, then time-series cluster analysis was used to identify the temporal evolution patterns of CLUE in various districts and counties. Furthermore, we explored the spatiotemporal dynamic relationship between CLUE and landscape pattern changes using landscape pattern index and geographically and temporally weighted regression (GTWR), considering spatiotemporal heterogeneity, and using interaction detectors to identify the interaction between landscape pattern factors on CLUE. The results show that: (1) From 2000 to 2020, CLUE in Henan Province varied between 0.50 and 0.70 in most years, indicating potential for improvement. There are four primary temporal evolution patterns: 26 Late-development, 22 Wave-rising, 27 Fluctuation-rising, and 29 Continuous-rising types of CLUE. (2) CLUE exhibits low values in the middle and eastern regions, i.e., areas with high values are concentrated in the southern districts, counties, and western and northern regions. CLUE has a significant positive spatial correlation with HH agglomeration areas mainly concentrated in Xinyang City, and LL agglomeration areas mainly located in the eastern and central regions. (3) Overall, different landscape factors exhibit varying degrees of spatiotemporal heterogeneity in their impact on CLUE. The total area and aggregation of cultivated land have a positive effect on CLUE, with the area of the positive influence of the total area gradually expanding over time and the aggregation gradually decreasing. The complexity of cultivated land shape has a negative effect. The impact of cultivated land patch density is two-sided, with the area of negative influence gradually expanding over time. (4) The interaction between the total area, shape, and aggregation of cultivated land is enhanced. Additionally, the interaction between cultivated land patch density and other factors has changed from a weakening to a strengthening one, and the "double-edged sword" effect has gradually shifted into a one-way effect. Therefore, in the process of land consolidation, it is recommended to prioritize regularized, larger, and more concentrated cropland patches whenever possible.

**Keywords:** SBM; GML index; GTWR; interaction detector; Mfuzz; spatial autocorrelation

## 1. Introduction

In the context of limited cultivated land, increasing grain production is an essential task for ensuring food security in China [1]. Rapid urbanization and economic growth over the past two decades have resulted in significant losses of cultivated land or its conversion into non-agricultural uses, causing a gradual reduction in cultivated land resources. The

intensive use of fertilizers and pesticides has led to a decline in land productivity, thereby seriously endangering the production environment of cultivated land. Serious problems, including non-agriculturalization, non-grain cultivation, and quality degradation [2], have negatively impacted food security. As a result, improving cultivated land use eco-efficiency (CLUE) and reducing negative ecological impacts during the production process have become phasing issues.

Previous research may have underestimated the actual efficiency values of CLUE. CLUE is a measure of the coordination between land use and the ecological environment of a region [3]. Measuring methods include ecological footprint [4,5], principal component analysis [6,7] (PCA), and data envelopment analysis (DEA) [8–11]. The super-efficiency SBM model [12] derived from DEA has become the mainstream model for measuring CLUE. It effectively solves the problem of slackness between input and output variables, reflecting the differences in efficiency boundaries among research units, without the need to specify the model's specific form or estimate parameters. Researchers have evaluated CLUE based on the SBM model using carbon emissions as an unexpected output [2,10], and the results show varying degrees of decline in CLUE in most provinces after the carbon emissions index is incorporated into the measurement system [13]. However, these studies only consider negative environmental impacts of agricultural production, such as carbon emissions [14,15], environmental pollution [16], or both [17], as unexpected outputs in the measurement system. Therefore, the positive environmental impacts of agricultural production, such as absorption and fixation of carbon dioxide in soil and plant biomass, may be overlooked, leading to an underestimation of the actual CLUE.

Ignoring temporal or local non-stationarity of CLUE, a regression analysis may produce inaccurate results. In a study of spatiotemporal evolution and its correlated influencing factors, Hou. et al. [18] employed a systematic generalized method to quantify urbanization's effects on CLUE. The results indicate that urbanization's agglomeration and barrier effects negatively impact CLUE. Using GWR, Ma. et al. [5] identified significant factors affecting CLUE, such as resource endowment, economic level, natural characteristics, and production conditions. The impact of natural endowments, socioeconomic factors, and agricultural productivity technology on CLUE in Guangdong province was analyzed by Zang et al. [2] using geographic detectors. Using a GWR model, Zhao et al. [10] observed a negative correlation between CLUE and elevation, slope, percentage of the agricultural population, and percentage of non-food crops. In general, exploration of variables in CLUE frequently makes use of global regression or GWR which overlooks local non-stationarity and temporal changing variables that may promote biased results.

Landscape pattern refers to the spatial distribution, structure, and configuration of land use and cover with various sizes, shapes, and attributes [19]. It is considered the most visually apparent representation of land use and cover [20]. The landscape pattern index [21] is focused on evaluating the impact of changes in the configuration and composition of land elements on ecological processes [22] and is an important indicator of landscape heterogeneity [23]. Changes in China's landscape pattern are primarily driven by alterations in land patches' size, shape, and attributes, particularly due to the reduction of cultivated land and spatial transfer. Economic development and urbanization contribute to the conversion of cultivated land to non-agricultural uses, resulting in cultivated land loss. The implementation of policies promoting a balance between land occupation and compensation has facilitated the transfer of cultivated land from developed regions to less developed ones [1]. Although China's cultivated land has experienced significant changes in its landscape pattern, few studies have explored the dynamic relationship between landscape pattern and CLUE.

The study aims to accurately reveal the spatial and temporal characteristics of CLUE in Henan Province, as well as to explore the dynamic relationship between the landscape pattern of cultivated land and CLUE. Henan Province holds a prominent position in China as a major grain producer and the largest wheat-producing province. Unfortunately, the excessive use of chemical fertilizers and pesticides has led to severe ecological problems in

crop production within Henan Province [24], resulting in a relatively low ranking for CLUE among China's provinces [25]. According to the data from 2019, Henan Province was the third-highest contributor to the national agricultural carbon emissions [26]. This poses a threat to China's food security and green sustainable development. The rapid urbanization process has contributed to a reduction in cultivated land area in Henan Province, amounting to a decrease of 5609.93 km$^2$ (22.78%) between 2000 and 2020 [27]. Concurrently, a transfer of cultivated land from developed areas to underdeveloped areas has occurred [28]. During the 13th Five-Year Plan period, Henan Province managed to supplement 2.1 million mu of cultivated land while occupying 1.16 million mu for construction purposes. The scale of spatial transfer of cultivated land is quite extensive. Despite the implementation of strict control measures in Henan Province, the phenomenon of "taking advantage of advantages and compensating for disadvantages" persists. CLUE in Henan Province has consistently ranked among the lowest in inter-provincial comparisons over the past two decades. In the past two decades, how has CLUE developed in each region of the province, and what are the characteristics of temporal and spatial evolution? The area, shape, and attributes of cultivated land patches change drastically. Does the change in the landscape pattern of cultivated land promote the change of CLUE? What is the dynamic relationship between the two? Is there spatial heterogeneity? Relevant research fields have not yet answered these questions.

Based on the aforementioned points, this study focuses on Henan Province at the county-level unit as the research scope. The objectives are as follows:

(1) Taking into consideration the impact of carbon emissions and carbon storage, the study aims to accurately depict the temporal and spatial evolution characteristics of CLUE in the county-level units of Henan Province;

(2) By utilizing GTWR, which accounts for spatiotemporal non-stationarity, the study aims to comprehensively explore the dynamic spatiotemporal relationship between the landscape pattern of cultivated land and CLUE;

(3) Through the use of the interaction detector, the study aims to analyze the driving effect of the interaction between factors related to the landscape pattern of cultivated land on CLUE.

This study has three major contributions. First, it investigates the dynamic relationship between changes in the landscape pattern of cultivated land and CLUE for the first time. This provides a theoretical framework for enhancing CLUE and implementing policies to safeguard cultivated land resources when facing limitations. Second, by considering both carbon storage and carbon emission effects, a more accurate description of CLUE can be achieved. This fosters a better understanding of CLUE under the influence of carbon effects, promoting the transition towards environmentally friendly production practices. Finally, the utilization of time series clustering identifies the development pattern of CLUE in county-level units within Henan Province. This perspective offers new insights into regional agricultural production management and the advancement of CLUE.

## 2. Materials and Methods

The study's flowchart is presented in Figure 1. Section 2 provides an introduction to the study area, data sources, and research methodology. Section 3 presents the econometric results of the CLUE evaluation and regression models. Section 4 discusses and analyzes these findings. Finally, Section 5 summarizes the main conclusions derived from this study.

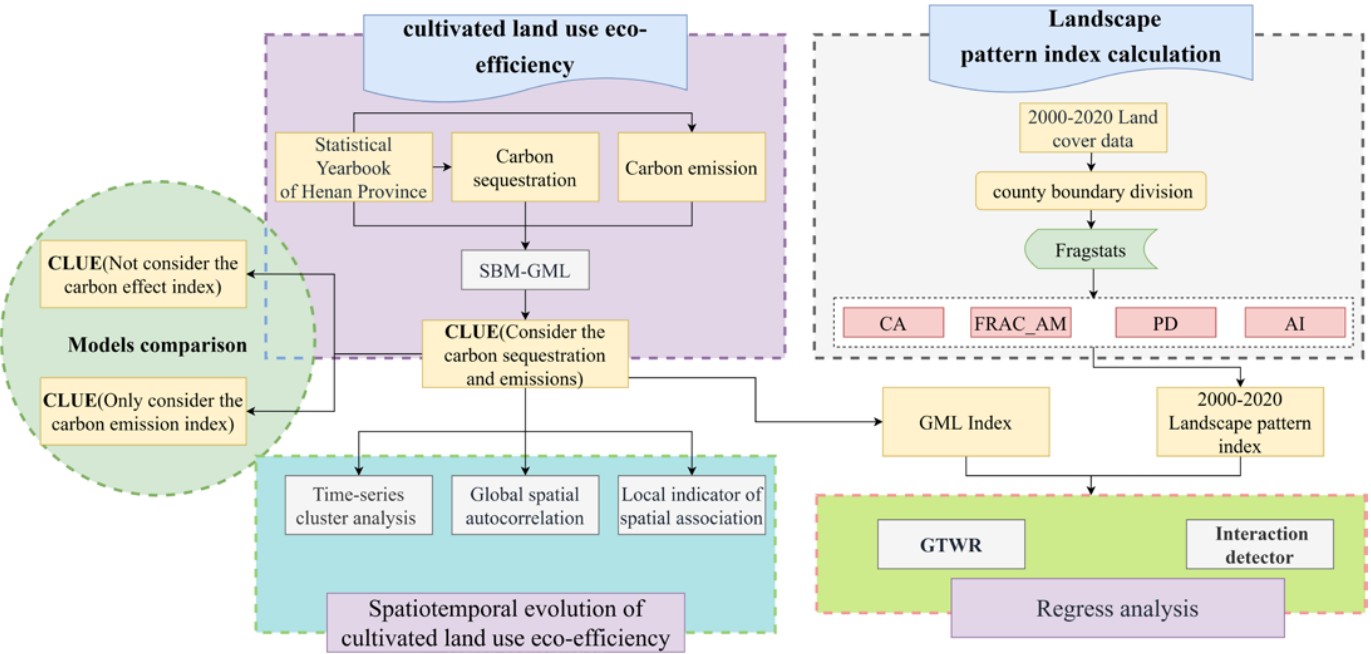

**Figure 1.** The flow chart.

### 2.1. Study Area

Located in the middle and lower reaches of the Yellow River Basin, Henan Province covers an area of 167,000 square kilometers, including 17 prefecture-level cities, which account for 1.73% of China's total landmass. As the primary grain production area and the second-largest grain-producing province in China, Henan Province is the country's highest wheat-producing province, thanks to its advantageous conditions such as soil quality and climate. Henan province's total cultivated land area, which is stable at more than 112 million mu, ranks third among all provinces in China. In 2020, the sown area of crops in China accounted for 220 million hectares or 8.78% of the total area, which played a crucial role in maintaining national food security and facilitating carbon cycling processes in ecosystems. Moreover, there were notable regional variations across the province, and Sanmenxia emerged as the city with the most productive CLUE at the municipal level [29]. However, county-level research on the matter has been insufficient. Carrying out research at a more microscopic level can not only account for socioeconomic disparities between various regions but also ascertain a diversified range of factors that drive CLUE. Thus, examining CLUE in Henan Province through a county-level analysis is imperative. In this study, we excluded areas within urban districts and county-level units that lacked adequate data due to administrative reorganization. The research area was ultimately determined to consist of 104 counties, as shown in Figure 2.

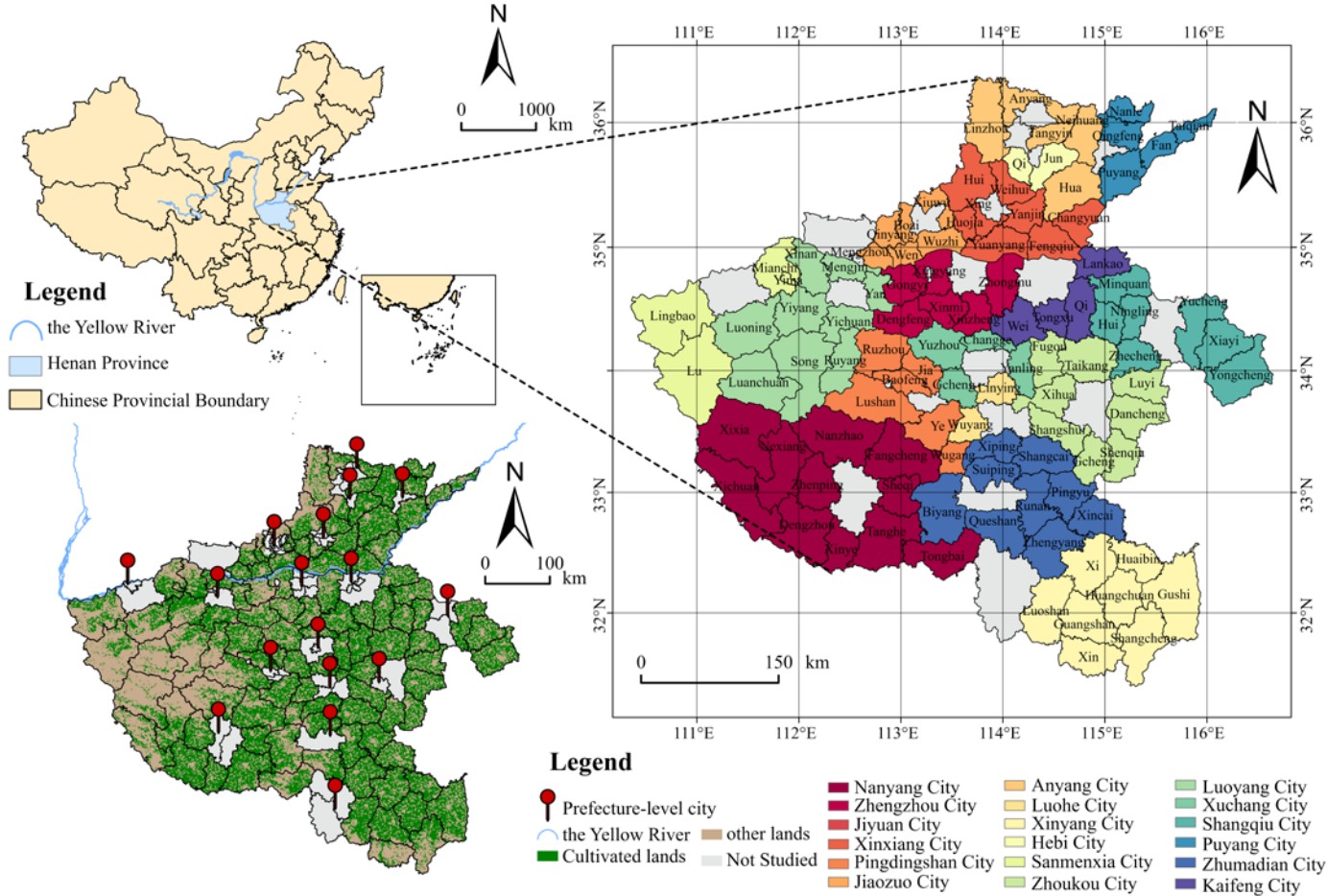

**Figure 2.** Spatial distribution of the study area.

*2.2. Data Sources*

Data used in the super-efficiency SBM model are sourced from the Statistical Yearbook of Henan Province. Missing data are filled using both social statistical bulletins and multi-imputation methods gathered from different counties and cities. Using the cultivated land use data derived from remote sensing images of land use by the Chinese Academy of Sciences Resource and Environment Science Data Center (www.resdc.cn, accessed on 21 June 2022), the study uses Fragstats 4.2 to calculate landscape pattern indices. The data cover the years 2000, 2005, 2010, 2015, and 2020, and has a spatial resolution of 1 km × 1 km.

*2.3. Measurement of Carbon Effects*

2.3.1. Carbon Sequestration

Crops undergo photosynthesis to absorb carbon dioxide from the atmosphere, which is then stored in the form of carbon in living organisms such as seeds, stems, leaves, roots, and soil. Carbon sequestration in agriculture refers to the ability of crops to absorb carbon dioxide from the atmosphere. Tian Yun et al. [30] and Han Zhaoying et al. [31] have developed validated methods for calculating carbon sequestration and emissions in agricultural production, which have been supported by various studies [32,33] and are suitable for research at the county level in Henan Province. The carbon sequestration calculation formula is as follows:

$$C = \sum_{i}^{K} C_i = \sum_{i}^{K} ca_i \times Y_i \times \frac{1-r}{HI_i} \tag{1}$$

The formula for determining carbon sequestration in a specific county in Henan Province is as follows: $C$ represents the overall carbon sequestration of the county, while $C_i$ represents the carbon sequestration of a specific crop. $K$ represents the crop variety, $ca_i$ represents the amount of carbon absorbed by crops through photosynthesis to synthesize unit organic compounds, $Y_i$ represents the economic output of crops, $r$ represents the water content of the crop's economic yield, and $HI_i$ represents the crop's economic coefficient. Table 1—Calculated parameters of carbon effect in Henan Province—displays the carbon absorption rate, water content, and economic coefficient [30,31] of the main crops in Henan Province.

**Table 1.** Calculated parameters of carbon effect in Henan Province.

| Index | Crop Variety | Carbon Absorption Rate | Economic Coefficient | Moisture Content |
|---|---|---|---|---|
| | Wheat | 0.485 | 0.40 | 0.12 |
| | Corn | 0.471 | 0.40 | 0.13 |
| | Other grains | 0.414 | 0.45 | 0.12 |
| Carbon sequestration | Other food crops | 0.45 | 0.40 | 0.12 |
| | Beans | 0.45 | 0.25 | 0.10 |
| | Cotton | 0.45 | 0.10 | 0.08 |
| | Oil | 0.45 | 0.34 | 0.13 |
| | Fruit | 0.45 | 0.70 | 0.90 |
| | Carbon source | Carbon emission coefficient | | |
| | Fertilizer | 0.8956 kgC × kg$^{-1}$ | | |
| | Pesticide | 4.9341 kgC × kg$^{-1}$ | | |
| Carbon emission | Agricultural film | 5.18 kgC × kg$^{-1}$ | | |
| | Agricultural machinery process | Crop cultivation area × 16.4 kgC × kg$^{-1}$ + Total power of agricultural machinery × 0.18 kgC × kw$^{-1}$ [34] | | |
| | N$_2$O | Calculated based on [30] | | |

### 2.3.2. Carbon Emission

The carbon emissions considered in this study are those that are generated during crop production. These emissions come from two sources: the use of agricultural materials, which includes the carbon emissions from chemical fertilizers, pesticides, agricultural films, and agricultural machinery, and the $N_2O$ emissions from soil surface damage during crop planting. $CH_4$ and other greenhouse gases emitted during rice cultivation are not included due to the lack of available data. Based on this, a formula has been constructed for calculating the carbon emissions from cultivated land use.

$$CE = \sum CE_i = \sum T_i \times \delta_i \qquad (2)$$

In the formula: $CE$ represents the carbon emissions from cultivated land use, $CE_i$ represents the carbon emissions from the $i$-th carbon source, $T_i$ represents the consumption of the $i$-th carbon source, and $\delta_i$ represents the emission coefficient of each carbon emission source. The coefficient for carbon emission sources in Henan Province [30] is displayed in Table 1. The model used in the calculation of carbon emissions converts $CO_2$ and $N_2O$ into standard carbon, according to the IPCC Fourth Assessment Report (2007).

### 2.4. CLUE Measure Based on Super-Efficiency SBM Model

The super-efficiency SBM model that considers undesired output effectively addresses the issue of slack variables and undesired output during the input and output phases. Moreover, it resolves the issue of comparing research units that have an efficiency value of 1. Considering that the efficiency values vary under different assumptions of constant

returns to scale (CRS) and variable returns to scale (VRS), the suggestion of Zheng [35] is followed and the results obtained under the assumption of VRS are given a higher priority. This model is utilized to measure and evaluate the CLUE in Henan Province. The calculation formula for this is presented as follows [9]:

$$\rho^* = \min \frac{\frac{1}{m} \sum_{i=1}^{m} \frac{\overline{x}}{x_{ik}}}{\frac{1}{s_1+s_2} \left( \sum_{s=1}^{s_1} \frac{\overline{y}^g}{y_{sk}^g} + \sum_{q=1}^{s_2} \frac{\overline{y}^b}{y_{qk}^b} \right)} \tag{3}$$

$$\overline{x} \geqslant \sum_{j=1,\neq k}^{n} x_{ij}\lambda_j; \ \overline{y}^g \leqslant \sum_{j=1,\neq k}^{n} y_{sj}^g \lambda_j; \ \overline{y}^g \geqslant \sum_{j=1,\neq k}^{n} y_{ij}^g \lambda_j; \ \overline{x} \geqslant x_k; \ \overline{y}^g \leqslant y_k^g; \ \overline{y}^b \geqslant y_k^b;$$

$$\lambda_j \geqslant 0, i = 1, 2, \cdots, m; \ j = 1, 2, \cdots, n; \ s = 1, 2, \cdots, s_1; \ q = 1, 2, \cdots, s_2.$$

The value of CLUE for each district and county in Henan Province is represented by $\rho^*$. $x$, $y_g$, and $y_b$ denote the input matrix, expected output, and non-expected output matrix of cultivated land production factors in the research unit, respectively. $n$ represents the number of research units, while $m$, $s_1$, and $s_2$ indicate, in order, the number of production factor inputs during cultivated land utilization, the number of expected output indicators, and the number of unexpected output indicators. The variables $\overline{x}$, $\overline{y_g}$, and $\overline{y_b}$ denote the redundancy of input, expected output, and unexpected output, respectively. The weight vector is represented by $\lambda$.

The CLUE is measured based on input and output during cultivated land utilization [9]. The input factors of cultivated land mainly comprise land, labor, and various materials used in production. The labor force input is measured by the agricultural labor force, whereas pesticide, agricultural film, chemical fertilizer usage, and the total power of agricultural machinery are measured for the input of material production materials. The expected output index refers to the economic and ecological effects of cultivated land utilization, including its economic value and carbon sequestration. Carbon emissions are used as an indicator of undesired output. All indicators are measured by the "average sown area of crops" [36], which provides a more accurate representation of the cultivated land utilization scenario. The indicators are explained in Table 2.

**Table 2.** CLUE evaluation index system.

| Variable | Indicators | Description | Unit |
|---|---|---|---|
| Input | Labor input | Agricultural labor force per unit of sown area | person/ha |
| | Pesticide input | Pesticide dosage per unit sown area | ton/ha |
| | Agricultural film input | Amount of agricultural film per unit sowing area | ton/ha |
| | Fertilizer input | Fertilizer application per unit sowing area | ton/ha |
| | Total power of agricultural machinery | Total power of agricultural machinery per unit sowing area | 10,000 kWh/ha |
| Expected output | Crop yield | Crop yield per unit sown area | ton/ha |
| | Gross Agricultural Production | Gross agricultural production per unit sown area | ten thousand yuan/ha |
| | Carbon sink | Carbon sequestration per unit planting area | ton/ha |
| Undesired output | Carbon emission | Carbon emissions per unit sown area | ton/ha |

### 2.5. Spatial Autocorrelation

Spatial autocorrelation analysis at global and local scales can reveal the spatial relationship between geographical units. The global spatial autocorrelation captures the attributes of spatial units throughout the study area and indicates the degree of similarity between the attributes of a given unit and its neighboring units [37], measured by using Moran's I index, a statistic that ranges from −1 to 1. Local autocorrelation analysis gauges the impact of neighboring spatial units on the overall spatial autocorrelation of the study region, in terms of the correlation between the ecological quality of the study unit and that of its neighboring units. The outcomes of Local Moran's I can be visualized by drawing a spatial association (LISA) map, in which HH and LL indicate that high-value (or low-value) areas are adjacent to their like counterparts, thereby showing positive spatial correlation whilst HL and LH signify that high-value (or low-value) areas are adjacent to their unlike counterparts, thus exhibiting negative spatial correlation.

$$Global\,Moran's\,I = \frac{n\sum_{i=1}^{n}\sum_{j=1}^{n}w_{ij}(x_i-\overline{x})(x_j-\overline{x})}{\sum_{i=1}^{n}\sum_{j=1}^{n}w_{ij}\sum_{i=1}^{n}(x_i-\overline{x})^2} \tag{4}$$

$$Local\,Moran's\,I = \frac{n(x_i-\overline{x})\sum_{j=1}^{m}W_{ij}(x_j-\overline{x})}{\sum_{i=1}^{n}(x_i-\overline{x})^2}, \ (i \neq j) \tag{5}$$

The formula includes several parameters: $n$ represents the number of research units, $x_i$ denotes the observed value, $w_{ij}$ denotes a spatial weight matrix connecting the $i$-th and $j$-th samples, and $\overline{x}$ is the mean value. Local spatial autocorrelation employs the standardized statistic Z to examine the significance of the spatial autocorrelation captured by Moran's $I$ index. The significance level for the study was set at 0.05.

### 2.6. Time-Series Cluster Analysis

The study employs the "Mfuzz" package in the R programming language [38] to perform a time-series cluster analysis of the CLUE. The method used is the fuzzy C-means (FCM) clustering algorithm, which has found extensive application in biological information and gene expression research fields [39]. The FCM clustering algorithm is an improved and fuzzy version of the K-means algorithm that applies a membership function for classification and determines the evolution pattern of each district and county based on the membership degree size. FCM enhances the algorithm's capacity to manage ambiguity and uncertainty over K-means. The objective function is established as follows [40]:

$$J(MC) = \sum_{i=1}^{n}\sum \mu_{ij}^{\varphi}d_{ij}^2$$
$$i = 1,\cdots,n; \ j = 1,\cdots,c \tag{6}$$

$C$ is the centroid matrix of the cluster category, $M$ is the fuzzy membership degree matrix $n \times c$ of CLUE in districts and counties, $\mu_{ij}$ ($\mu_{ij} \in [0,1]$) is the membership degree value of the $i$-th district and county corresponding to the $j$-th cluster, and $\varphi(\varphi \geq 1)$ is the fuzzy weighting index. $d_{ij}^2$ is the square of the distance between the $i$-th district and county corresponding to the $j$-th category center.

### 2.7. GTWR

The super-efficiency SBM model can only measure the relative CLUE among research units in the same period. To analyze the dynamic changes of efficiency over time, the SBM-GML index is introduced due to its time series comparability. Consequently, this

study employs the GML index to dynamically evaluate CLUE in Henan Province. The specific calculation formula for the GML index is presented [41,42] as follows:

$$
\begin{aligned}
\mathrm{GML}^{t,t+1}&\left(x^t, y^t, b^t, x^{t+1}, y^{t+1}, b^{t+1}\right) = \\
&\frac{D_v^{t+1}\left(x^{t+1}, y^{t+1}, b^{t+1}\right)}{D_v^t\left(x^t, y^t, b^t\right)} \times \left(\frac{D_c^t\left(x^t, y^t, b^t\right)}{D_c^{t+1}\left(x^t, y^t, b^t\right)} \times \frac{D_c^t\left(x^{t+1}, y^{t+1}, b^{t+1}\right)}{D_c^{t+1}\left(x^{t+1}, y^{t+1}, b^{t+1}\right)}\right)^{1/2} \\
&\times \frac{D_c^{t+1}\left(x^{t+1}, y^{t+1}, b^{t+1}\right)/D_v^{t+1}\left(x^{t+1}, y^{t+1}, b^{t+1}\right)}{D_c^t\left(x^t, y^t, b^t\right)/D_v^t\left(x^t, y^t, b^t\right)}
\end{aligned}
\tag{7}
$$

In the formula: $x$ is the input factor, $y$ is the expected output, $b$ is the non-radial expected output, and $t$ is the year. Among them, $D(x_t, y_t, b_t)$ is a directional distance function, and $D_c$ and $D_b$ are directional distance functions based on constant returns to scale and variable returns to scale, respectively.

To eliminate the spatiotemporal non-stationarity result bias, the GTWR model embeds the theme of time into the geographically weighted regression (GWR) model, creating a more efficient parameter estimation. The study employed the GTWR plug-in operation model in ArcGIS developed by Huang et al. [43] to investigate the spatiotemporal dynamic correlation between CLUE and landscape pattern changes in Henan Province from 2000 to 2020. The model is constructed as follows:

$$
Y_{\mathrm{i}} = \beta_0\left(X_i^t, Y_i^t, T_i\right) + \sum_k \beta_k\left(X_i^t, Y_i^t, T_i\right) X_{ik} + \varepsilon_i
\tag{8}
$$

$Y_i$ is the county-level CLUE-GML index of Henan Province, $(X_i^t, Y_i^t, T_i)$ is the space-time coordinates of the $i$-th district and county, $\beta_0$ (the constant term of point $i$ in the GTWR model, $\beta_k$) is the $k$-th point of $i$ point A regression parameter;—that is, the weight of the model function at the time and space coordinates $(X_i^t, Y_i^t, T_i)$. $X_{ik}$ is the value of the land use landscape pattern index $X_k$ at point $i$ compared with the first period—that is, the change rate of each quantitative index of the landscape pattern.

### 2.8. Interaction Detector

The geographic detector method can detect the spatial distribution characteristics of geographic elements or geographic phenomena [44], and the interaction detector is used to identify the interaction between two driving factors. That is to analyze whether the driving force for the change of CLUE is enhanced or weakened when the landscape pattern variable *X*1 and the landscape pattern variable *X*2 act together. Table 3 presents the types of interaction between each pair of variables.

**Table 3.** Interaction Type Criteria.

| Criterion | Interaction |
|---|---|
| $q(X1\cap X2) > q(X1) + q(X2)$ | Non-linear enhancement |
| $q(X1\cap X2) > \mathrm{Max}[q(X1), q(X2)]$ | Two-factor enhancement |
| $q(X1\cap X2) = q(X1) + q(X2)$ | Independent |
| $q(X1\cap X2) < \mathrm{Min}[q(X1), q(X2)]$ | Non-linear attenuation |
| $\mathrm{Min}[q(X1), q(X2)] < q(X1\cap X2) < \mathrm{Max}[q(X1), q(X2)]$ | One-factor nonlinear attenuation |

## 3. Results

### 3.1. Measure CLUE and Model Comparison

After estimating the carbon sinks and carbon emissions of each region from 2000 to 2020 using Formulae (1) and (2), the SBM model is used to calculate the CLUE taking into account the combined effect of carbon emissions and carbon sequestration (Model.1), and three measurements were compared. Model.2 and Model.3, respectively, use the same model as Model.1 to calculate CLUE without considering the carbon effect index and only considering the carbon emission index. Figure 3 shows that the mean and median of Model.2 and Model.3 are lower than those of Model.1 each year, which means that the

carbon effect has a positive impact on the CLUE, and only considering the negative impact of carbon emissions will underestimate the real CLUE.

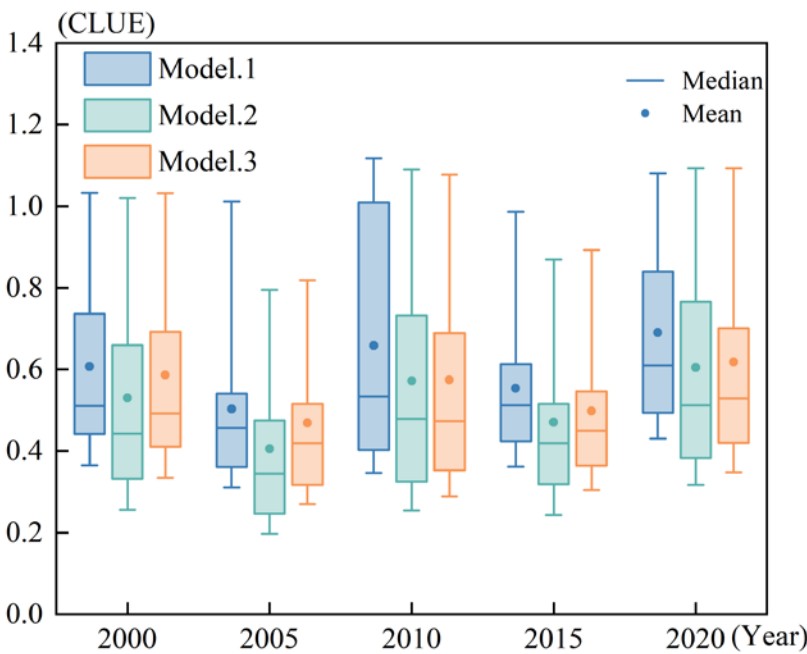

**Figure 3.** Model comparison.

The results show that the average value of CLUE in Henan Province from 2000 to 2020 showed a trend of increasing fluctuations, and the value in most years was mainly between 0.50 and 0.70, indicating that CLUE in most counties in Henan is not high, and there is still great potential for growth in the future. In terms of the time change, the average value of CLUE dropped slowly from 0.61 in 2000 to the lowest point of 0.50 in 2005, then rose to 0.65 in 2010, then dropped to 0.55 in 2015, and the rose rapidly to 0.69 in 2020, reaching the research period highest value within. Compared with the previous ten years, the growth rate of CLUE in the next ten years has slowed down.

*3.2. Spatial and Temporal Characteristics of CLUE*

According to the relevant research [1,10] and calculation results, CLUE is categorized into four categories: low efficiency (0, 0.6), medium efficiency [0.6, 0.8), medium and high efficiency [0.8, 1), and high efficiency (1, +∞). Figure 4 displays the spatiotemporal pattern of CLUE across 104 counties in Henan Province from 2000 to 2020. Generally, the spatial pattern of CLUE in Henan Province is low in the middle and east, while high-value areas are more concentrated in southern districts and counties, and the west and north have performed well. In 2000, there were 18 counties with CLUE that exceeded 1.0, indicating that they have reached DEA's relative effectiveness and had high CLUE. These counties were mainly concentrated in southern and central Henan (mainly in Lushan County and Ye County), Yichuan and Luoning counties in western Henan, and Qixian and Xiuwu counties in northern Henan. Since then, the number of high-efficiency counties has declined to 11 in 2005; but since 2010, the number has increased, reaching 28. Simultaneously, their spatial distribution gradually shifted to the west and north, with high-efficiency counties appearing in contiguous regions within northern Henan. In 2015, the number of high-efficiency counties decreased to 10, and as a result, the CLUE in these areas has significantly dropped. In 2020, the number of high-efficiency counties in Henan Province increased to 25, and the eastern region has seen an increase in high-efficiency counties, mostly in Shangqiu City, Luyi County, Shangshui County, and Xiangcheng City in Zhoukou City.

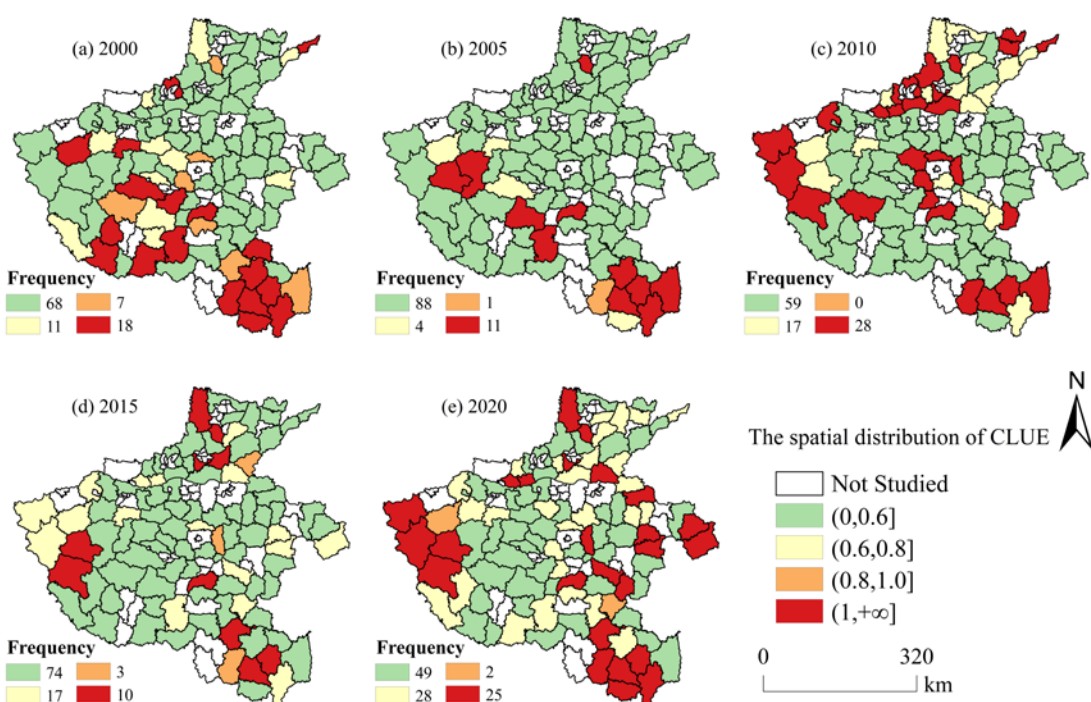

**Figure 4.** Spatial pattern of CLUE in Henan Province.

Using the R language Mfuzz package, we identified a dynamic evolutionary model of GML indices that reflects county-level CLUE in Henan Province from 2000 to 2020. We assessed the clustering effect using two methods, namely, Calinski–Harbasz and silhouette coefficient scores. Both metrics compare the variation between clusters and the variation within clusters (with the between-group variance being the largest and the within-group variance being the smallest). A higher score implies a better clustering effect. Based on Figure 5, we can observe that the dynamic evolutionary model of CLUE in Henan Province is categorized into four optimal types.

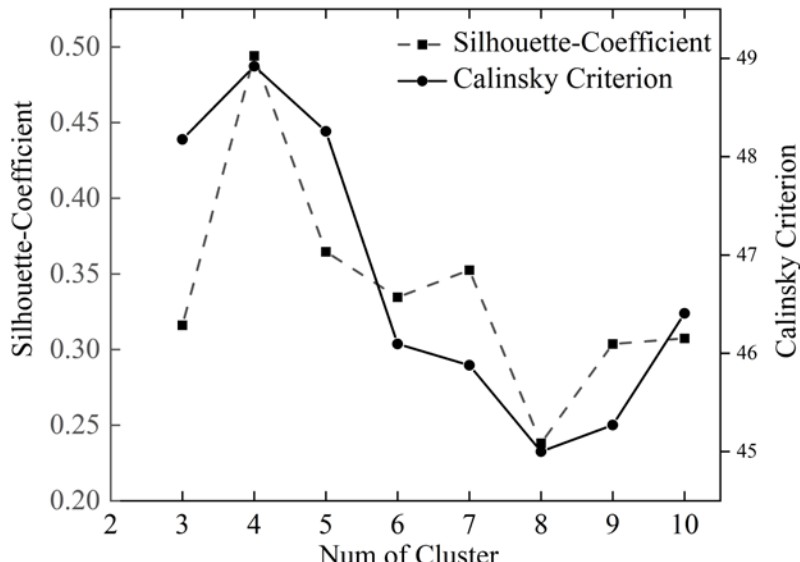

**Figure 5.** Selection of the optimal number of clusters.

Figure 6 displays the four CLUE development models from 2000 to 2020, with 26, 22, 27, and 29 counties identified within each cluster. Cluster 1, located mainly in Xinxiang City

and Jiaozuo City, is classified as a late-developing and upgrading district and county. There was a downward trend from 2000 to 2015, which can be attributed to the relatively strong industrial foundation in these areas, resulting in the absorption of a large amount of labor by industrial enterprises, and a lack of interest from farmers in cultivation. Nevertheless, the development of the "one village, one product" initiative has led to the formation of specialized villages focused on high economic yields in some areas, which has led to a rapid increase in CLUE. Cluster 2 is classified as a Wave-rising district and county. These counties are spatially dispersed, mainly found in Anyang and Sanmenxia cities. These regions have responded to the national policies promoting green agricultural development and ecological farming. As a result, they have accelerated the establishment of ecological farms and green-oriented agricultural modernization demonstration areas. This progress has contributed to the notable upward trend in CLUE. Cluster 3, primarily located in the southwestern part of Henan Province, is classified as a fluctuating rising district and county. Terrains in these areas are relatively complex and the cultivated lands are fragmented. Nonetheless, with an advance in traffic accessibility and planting technologies, including fertilization and irrigation, farmland production conditions have significantly improved. In addition, good ecological environments in these areas are conducive to the development of green agriculture. Cluster 4, mainly located in Shangqiu, Zhoukou, and Zhumadian in the eastern and southern parts of Henan Province, is classified as a continuously rising district and county which maintained rapid upward trends from 2005 to 2020. The Eastern Henan Plain is designated as an agricultural restricted development zone that plays a strategic role in the national agricultural product supply chain. Strong supportive policies and fast economic development have resulted in the improvement of cultivated land use efficiency, which is of significant importance for ensuring food production safety and environmental protection.

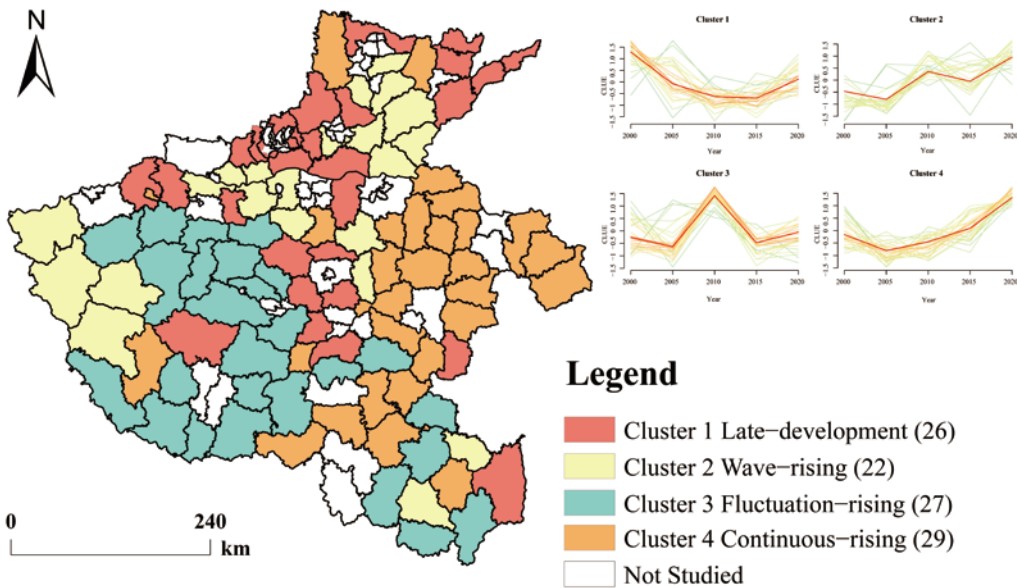

**Figure 6.** Spatial distribution of different evolutionary patterns. Note: The broken line in the figure is divided into green, yellow, and orange. The closer its color is to orange, the closer it represents the evolution trend of the CLUE in the district and county to the core features expressed by the cluster. Red is the centerline of the cluster.

### 3.3. Spatial Correlation Analysis

Based on the global Moran's I index, CLUE at the county level in Henan Province from 2000 to 2020 was examined. Table 4 reveals that the CLUE in Henan Province passed the 1% significance test and had a positive Moran's I value, indicating a significant positive space for CLUE at the county level. During the research period, Moran's I value demonstrated a

trend of rising initially, dropping, and then trending towards stability. From 2000 to 2005, spatial dependence for CLUE at the county level in Henan Province increased, and the spatial autocorrelation value decreased from 0.34 to 0.19 from 2005 to 2015, and slightly improved from 2015 to 2020. Overall, the spatial distribution pattern of CLUE at the county level in Henan Province is relatively stable and has not undergone major changes.

**Table 4.** Spatial autocorrelation analysis.

| Parameter\Year | 2000 | 2005 | 2010 | 2015 | 2020 |
|---|---|---|---|---|---|
| Moran's I | 0.29 | 0.34 | 0.24 | 0.19 | 0.20 |
| Z | 4.61 | 5.45 | 3.83 | 3.06 | 3.23 |
| P | 0.00 | 0.00 | 0.00 | 0.00 | 0.00 |

Based on the analysis of local spatial autocorrelation, the local spatial pattern of carbon efficiency can be further explored. According to Figure 7, the cluster types of CLUE in Henan Province are mainly HH clusters and LL clusters. From 2000 to 2020, the HH clusters were mostly concentrated in Xinyang City, indicating positive development for CLUE in this area. From 2000 to 2010, the LL clusters were predominantly in Zhengzhou and Kaifeng. From 2010 to 2020, the LL clusters shifted westward, forming an area of low CLUE radiating outward from Pingdingshan as the core.

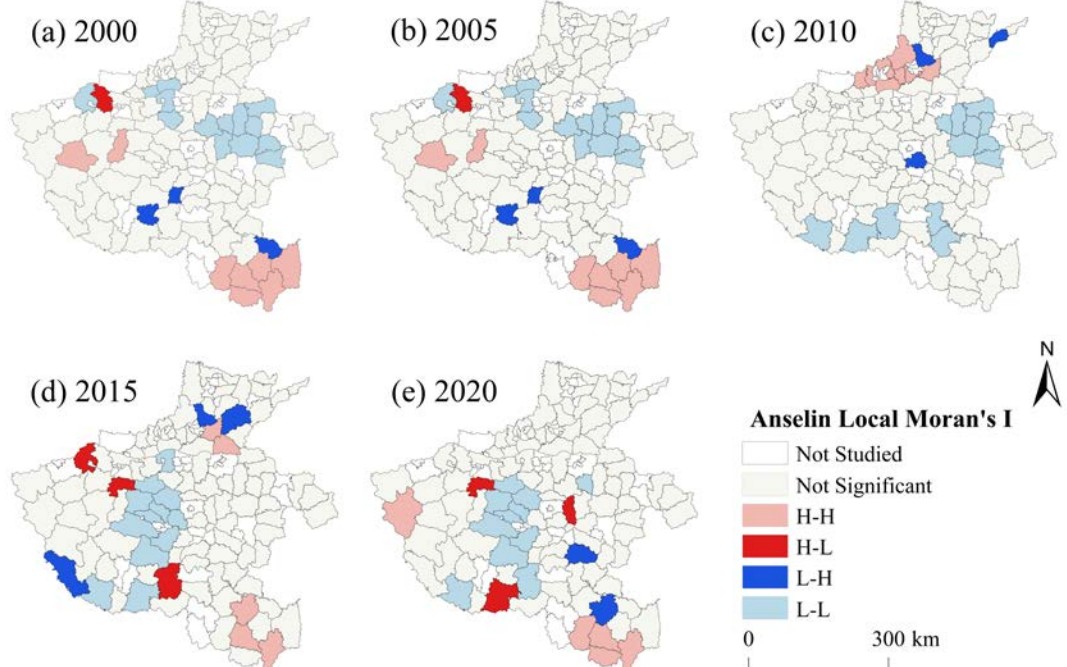

**Figure 7.** Local autocorrelation of CLUE in Henan Province.

In 2000, the L-H clusters were located within Nanyang City and Xinyang City, mainly on the outskirts of the HH agglomeration area. In 2005, Xin'an County and Mianchi County saw the emergence of HL and LH clusters, respectively. By 2010, HH agglomerations appeared in several counties of Xinxiang and Jiaozuo, suggesting an improvement in the CLUE in the region from 2005 to 2010. From 2010 to 2015, the LL clusters moved westward, and from 2015 to 2020, both H-L and LH agglomerations emerged. The spatial distribution of agglomeration areas in Henan Province remained relatively stable except in the northern region, indicating a tendency toward spatial dependence stability.

### 3.4. Quantitative Measurement and Dynamic Change of Landscape Pattern

The study computed the landscape pattern index by utilizing raster data of cultivated land from remote sensing images. Six frequently employed landscape pattern indicators were chosen at the Class level with the aid of Fragstats software (version 4.2) [45]. The indicators include CA, which serves as an essential indicator of cultivated land contraction and expansion by signifying the total area of cultivated land. Additionally, FRAC_AM was utilized to reveal the degree of shape complexity of plaques based on fractal geometry with values ranging from 1 for the simplest shapes to nearly 2 for more intricate perimeter shapes [46]. PD is the abbreviation for patch density, which represents the total number of patches in a region divided by the total area of cultivated land to reflect the fineness of the landscape. On the other hand, AI depicts the degree of aggregation of similar adjacent patches of cultivated land. The rationale for selecting these indicators is: (1) to select a reasonable combination of indicators to comprehensively describe the spatial pattern of cultivated land [1]; (2) to refer to previous studies and hand-pick highly recommended and reliable indicators [47]; and (3) to primarily consider comprehensible and computable landscape metrics while using metrics that are parsimonious and self-contained to reduce information redundancy [48].

Figure 8 demonstrates the changes in the landscape pattern index of each region in Henan Province from 2000 to 2020 with a 5-year interval. The total area of cultivated land continued to decrease during the study period. In the decade from 2015 to 2020, the pace of decline in agricultural land area in Henan Province notably increased. The FRAC_AM index persistently increased, signifying that the shape of cultivated land in each county had become more complex. The possible driving factor for this change was the transformation of numerous surrounding cultivated lands to other forms of land, primarily for construction. From 2000 to 2020, there was a steady increase in the PD value, indicating the gradual fragmentation of cultivated land in Henan Province, and the decline in the AI index confirmed this point.

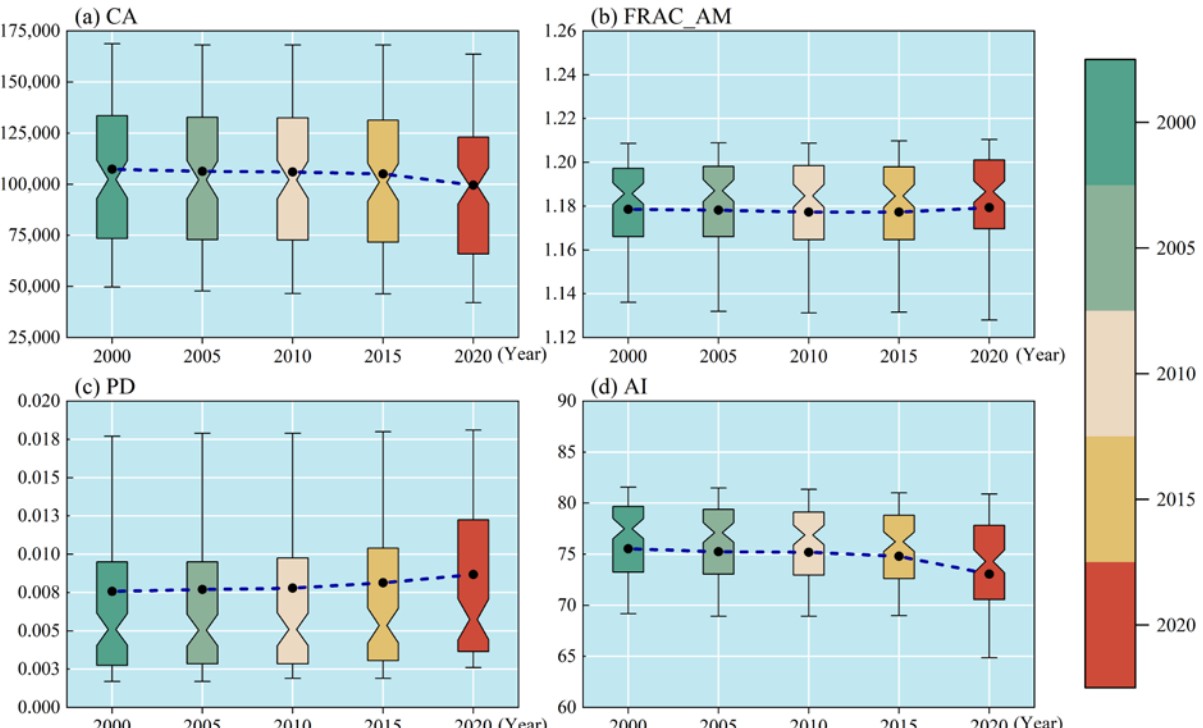

**Figure 8.** Dynamic changes in landscape patterns 2000–2020. Note: The connection point on the graph is the mean value, and the gap in the box is the median.

*3.5. GTWR Regression Analysis*

3.5.1. Regression Model Selection

Before conducting the regression analysis, the respective variables underwent a multicollinearity test. The results indicated that the expansion variance factors among the selected variables were all under 3, and there was no significant multicollinearity problem, which is in line with the research requirements. Further, spatiotemporal geographic weighted regression was performed, and the model fitting results were compared to those of OLS, TWR, and GWR models. As shown in Table 5, the AICc value of GTWR was lower than that of TWR and GWR, and $R^2$ was higher than that of TWR and GWR, demonstrating significant spatial heterogeneity in the CLUE in Henan Province in both time and space dimensions. The GTWR model considering the time-varying dynamic perspective is more advantageous for this study.

**Table 5.** Comparison of model selection.

| Parameter | OLS | TWR | GWR | GTWR |
|---|---|---|---|---|
| Residual Squares | 9.83 | 7.56 | 9.33 | 6.54 |
| Sigma | - | 0.13 | 0.15 | 0.13 |
| AICc | −367.29 | −441.71 | −347.75 | −460.68 |
| $R^2$ | 0.01 | 0.24 | 0.07 | 0.35 |
| $R^2$ Adjusted | - | 0.24 | 0.06 | 0.34 |

3.5.2. GTWR Results

Table 6 illustrates the effects of changes in landscape patterns on CLUE according to the GTWR model. Generally, an increase in CA and AI has a positive impact on CLUE, while FRAC_AM and PD have a negative impact. Additionally, based on the median coefficient of each factor, the degree of influence on CLUE is as follows: FRAC_AM, PD, AI, and CA, ranked in order of decreasing degree of impact.

**Table 6.** GTWR regression coefficients.

| Index | Average | Min | Median | Max |
|---|---|---|---|---|
| CA | 0.35 | −1.35 | 0.11 | 4.54 |
| FRAC_AM | −0.44 | −2.79 | −0.23 | −0.02 |
| PD | −0.44 | −3.13 | −0.33 | 1.50 |
| AI | 0.42 | −7.30 | 0.13 | 4.33 |

(1) The median and mean influence coefficients of the CA factor on CLUE are positive, with values of 0.11 and 0.35, respectively, indicating an overall positive performance of CA. According to Figure 9, the impact direction in most regions has changed from positive to negative and then back to positive, revealing strong spatial heterogeneity in the regression coefficient of CA. However, this spatial heterogeneity has weakened in the recent period. Between 2000 and 2005, the areas with negative impacts of CA were mainly located in the central and northern regions, with the intensity gradually weakening from north to south. The area of strong positive correlation was found in Xinyang City in the south of Henan Province, while the eastern and western areas showed moderate and weak positive correlations. From 2005 to 2010, CA and CLUE exhibited positive correlations in all regions of Henan Province, and the spatial heterogeneity decreased. The areas that were weakly positively correlated during this period mostly transformed from the previously negatively correlated areas. During the period from 2000 to 2010, the area of the positive influence of CA gradually expanded, and the positive effect strengthened. However, from 2010 to 2015, there was a sharp increase in the negative impact areas of CA, with only Lingbao City and Lushi County in Sanmenxia City showing a positive correlation, while the remaining areas turned mostly negative. The intensity of the negative impact

weakened gradually from northeast to southwest. With the rapid advancement of urbanization, the economic benefits of agriculture have fallen behind those of the industry and service sectors. Consequently, the expansion of cultivated land does not increase farmers' enthusiasm for agricultural production, and the shortage of agricultural labor, alongside the misuse of pesticides and chemical fertilizers, increases the cost of cultivated land management, resulting in a decrease in the CLUE. From 2015 to 2020, the influence of CA has mostly become positive, largely concentrated between 0.01 and 0.50, with a more noticeable spatial correlation. Only six districts and counties showed a weak negative correlation, indicating that the reduction in the cultivated land area would hinder the improvement of CLUE.

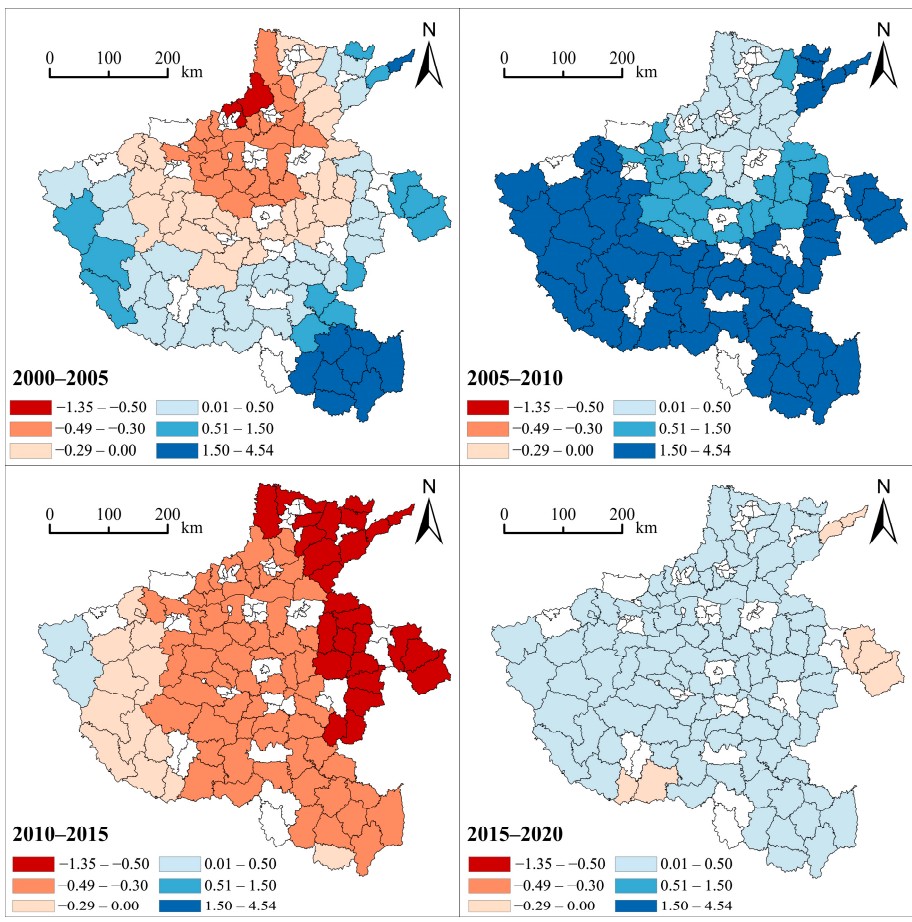

**Figure 9.** Spatial distribution of CA coefficients.

(2)    The median and mean values of the regression coefficients of FRAC_AM are −0.23 and −0.44, respectively, indicating that patch complexity hinders the improvement of CLUE. Compared with CA, FRAC_AM has a greater influence. Figure 10 shows that the influence of FRAC_AM has weak spatial heterogeneity, and the area of negative influence is widely distributed and strong. From 2000 to 2020, the response of county-level CLUE to the increase of FRAC_AM in Henan Province was always negatively correlated, indicating that the smoother and more regular the shape of the cultivated land patch, the faster the CLUE improvement. From 2000 to 2005, from the southwest to the northeast of Henan Province, the intensity of the negative influence gradually weakened, and most of the regional regression coefficients were between −1.49 and −0.15. From 2005 to 2010, the intensity of the negative impact in the north and south was weak, and the regression coefficient in the west was still relatively large, decreasing from west to east. From 2010 to 2015, the influence intensity of FRAC_AM weakened, and the spatial pattern was similar to that from 2005 to 2010. From 2015 to

2020, the spatial pattern of the regression coefficient of FRAC_AM decreased from the east and west sides to the middle. From 2000 to 2010, the negative driving force of patch shape complexity on CLUE gradually increased. From 2010 to 2020, the influence intensity of cultivated land patch complexity gradually decreased, mostly between −0.39 and 0.

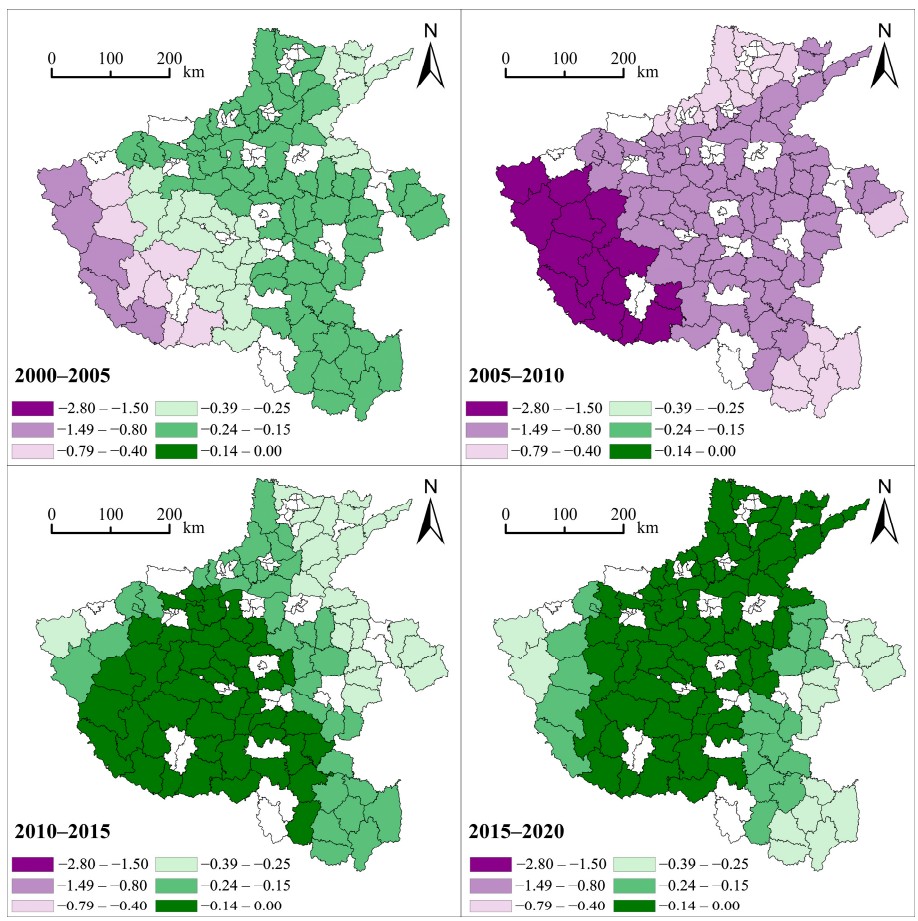

**Figure 10.** Spatial distribution of FRAC_AM coefficients.

(3)     The median and mean values of the regression coefficients for PD are −0.33 and −0.44, respectively. Compared to CA and FRAC_AM, PD has a stronger influence. Figure 11 shows that the regression coefficient of PD exhibits significant spatial heterogeneity, with alternating positive and negative effects observed in different regions. However, overall, an increase in the patch density of cultivated land impedes improvements in CLUE. From 2000 to 2005, the positive impact area lies below the axis, roughly with Xin'an County of Luoyang City and Gushi County of Xinyang City as the axis point, while the negative impact area is above the axis. The positive influence area exhibits a strong spatial correlation, and the strength of the negative influence area gradually increases with the distance from the axis. From 2005 to 2010, only two districts and counties show a positive correlation, while the rest exhibit a negative correlation, with the negative correlation strength decreasing from the eastern and western sides towards the middle. From 2010 to 2015, the strong negative correlation area disappears, with most areas showing a weak negative correlation between −0.59 and −0.1, and the positive impact area mainly concentrated in Xinyang City. From 2015 to 2020, all districts and counties display a negative correlation, and the spatial heterogeneity of regression coefficients diminishes, with the influence intensity of most districts and counties falling between −0.29 and −0.10.

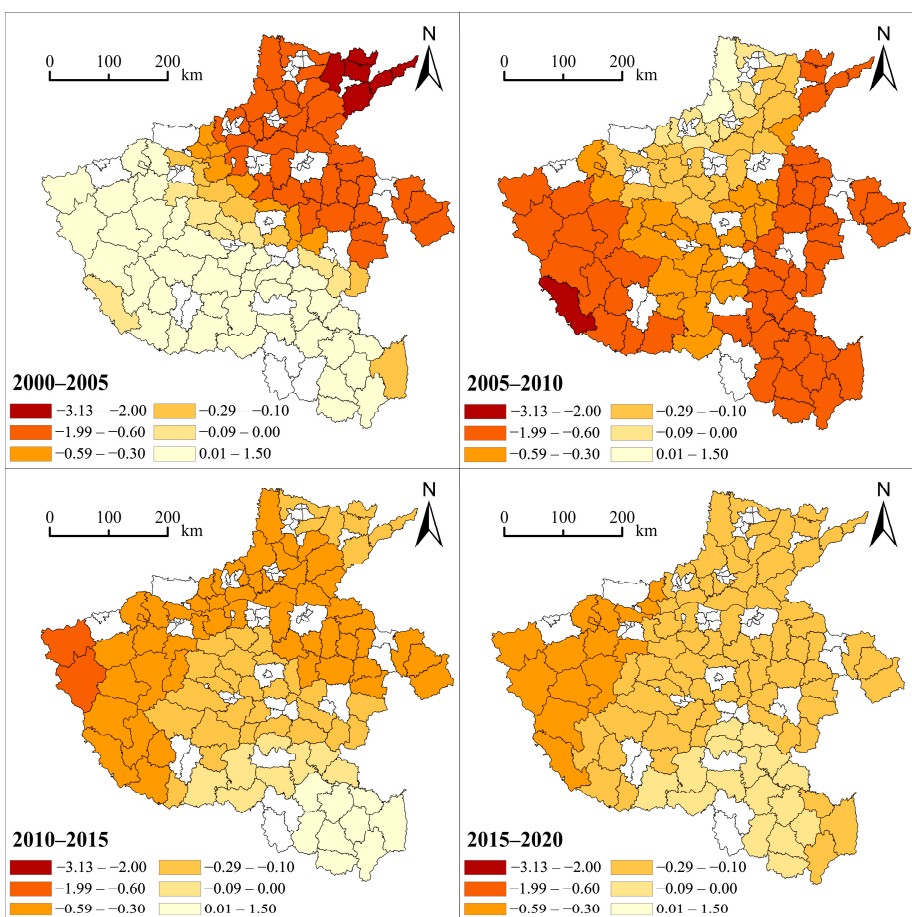

**Figure 11.** Spatial distribution of PD coefficients.

(4)     The median and mean regression coefficients for AI are 0.13 and 0.42, respectively. The influence of AI is greater than that of CA and less than that of FRAC_AM and PD. Figure 12 indicates strong spatial heterogeneity in the AI regression coefficient, with significant regional variations in positive and negative effects observed across different periods. However, overall, a denser concentration of cultivated land tends to contribute more to the improvement of CLUE. From 2000 to 2005, the positive impact area lies to the west of the central region, while the negative impact area lies to the east, with the regression coefficient gradually weakening as one moves from west to east. The south exhibits the strongest negative effect, while the west shows that AI improvement promotes the enhancement of CLUE. From 2005 to 2010, in the western region, the influence of AI gradually turns negative, with the boundary between positive and negative effects shifting from a north-south boundary to an east-west boundary, resulting in positive influence in the north and negative influence in the south. From 2010 to 2015, the number of negatively affected regions in the western region increases significantly, with the majority of remaining regions exhibiting a positive correlation. During this period, the regression coefficient shows strong spatial correlation and less pronounced spatial heterogeneity. From 2015 to 2020, only the northwest and southeast regions show a positive correlation, forming a high-value CLUE cluster where improved AI leads to a rapid increase in the CLUE. The majority of the remaining regions display a negative correlation with enhanced degrees, resulting in increased spatial heterogeneity. In the northern region, the impact of AI on improving CLUE shows signs of weakening.

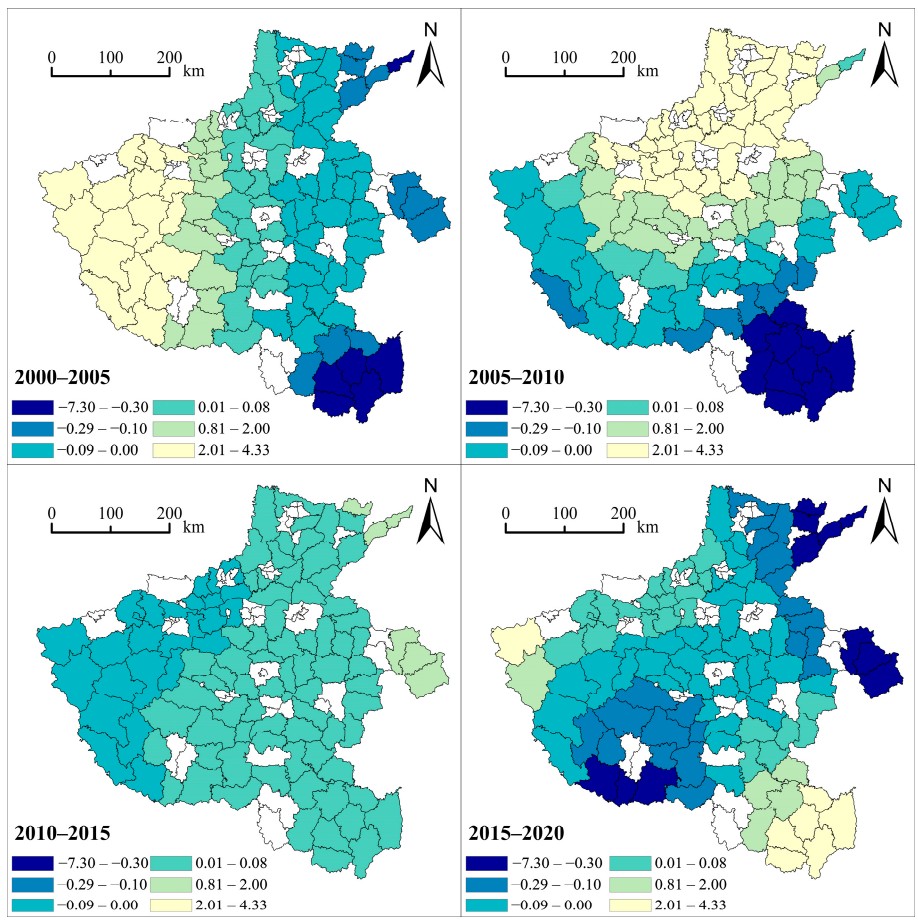

**Figure 12.** Spatial distribution of AI coefficients.

### 3.5.3. Robustness Check

In this study, we aim to evaluate the robustness of GTWR results in three aspects. Firstly, residual analysis is conducted to identify significant spatial correlation. When residuals exhibit such correlation, research outcomes cannot be trusted, so we perform a spatial autocorrelation analysis. The results confirmed that the residuals follow a normal distribution, as evidenced by the non-rejection of the null hypothesis at a 10% significance level with Z and P values of 0.11 and 0.91, respectively.

Second, to ensure the accuracy of our analysis, we performed variable substitution in Reg_Model.2 and Reg_Model.3. We replaced the FRAC_AM index in Reg_Model.2 with SHAPE_AM, which indicates the area-weighted average shape index determining the plaque's complexity based on Euclidean geometry. For example, when the plaque takes the shape of a square, its SHAPE_AM equals 1, while more irregular shapes result in an unlimited increase in the value. In Reg_Model.3, we replaced CA with GYRATE_AM, which represents the weighted average distance from the pixel center to the centroid within the patch to describe the cultivated land patch's area. According to Table 7, although Reg_Model.2 and Reg_Model.3 have lower $R^2$ and $R^2$ values compared to the Reg_Model.1, there were no significant changes in the size and direction of the coefficients. We employed supplementary variables in Reg_Model.4, where we incorporated the LSI index to indicate the landscape's shape index. LSI equals 1 for a landscape with only one square patch and increases as the landscape becomes more irregular. Results from our study confirmed that adding the LSI variable did not significantly affect the coefficients of explanatory variables or the diagnostic information, supporting the robustness of our regression model.

**Table 7.** Robustness Check.

| Parameter | Reg_Model.1 | Reg_Model.2 | Reg_Model.3 | Reg_Model.4 |
|---|---|---|---|---|
| CA | 0.11 | 0.03 | | 0.04 |
| GYRATE_AM | | | 0.19 | |
| FRAC_AM | −0.33 | | −0.38 | −0.22 |
| SHAPE_AM | | −0.14 | | |
| LSI | | | | −0.08 |
| PD | −0.23 | −0.23 | −0.42 | −0.33 |
| AI | 0.13 | 0.15 | 0.11 | 0.02 |
| Residual Squares | 6.54 | 6.94 | 6.83 | 6.64 |
| Sigma | 0.13 | 0.13 | 0.13 | 0.13 |
| AICc | −460.68 | −450.22 | −468.55 | −451.17 |
| $R^2$ | 0.35 | 0.31 | 0.32 | 0.34 |
| $R^2$ Adjusted | 0.34 | 0.30 | 0.31 | 0.33 |

Note: The coefficient value of the explanatory variable in the table is the median of the regression coefficient.

*3.6. Interaction Analysis*

Interactive detection can evaluate whether the pairwise effects of landscape pattern indicators contribute to improved explanations for changes in CLUE. Overall (Figure 13), there is substantial explanatory power in the interaction between FRAC_AM and other factors. Most interaction values are greater than the maximum value of a single factor, indicating that landscape pattern factors' impact on CLUE is not independent but synergistic.

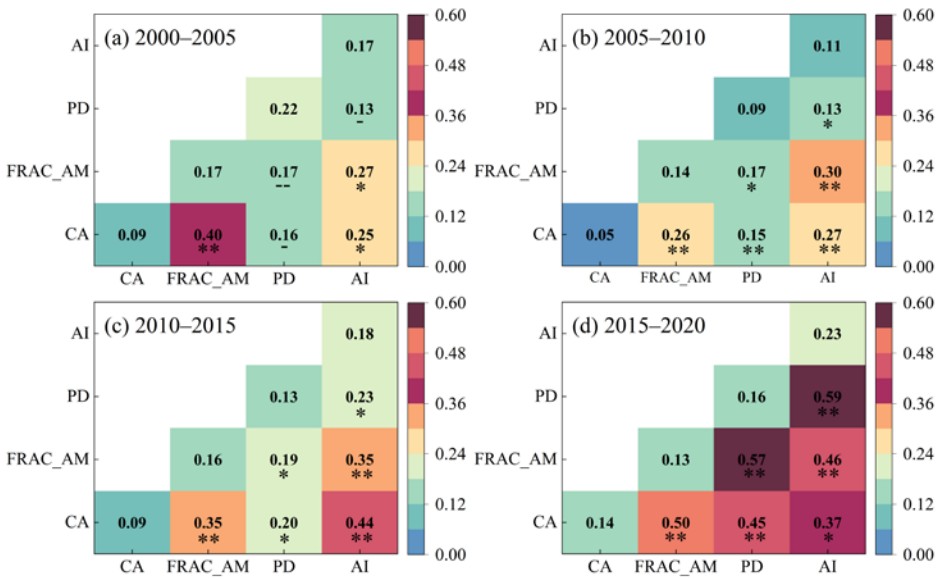

**Figure 13.** Interaction Heatmap. Note: "**" means non-linear enhancement, "*" means two-factor enhancement, "--" means single-factor nonlinear attenuation, and "-" means non-linear attenuation.

The interaction between CA and FRAC_AM consistently exhibits nonlinear enhancement, with the explanatory power showing a fluctuating upward trend. From 2000 to 2020, the interaction is consistently significant and relatively strong. The interaction between CA and PD initially weakens in the period 2000–2005, but from 2005–2020, it consistently demonstrates enhancement. Among these periods, 2005–2010 and 2015–2020 show nonlinear enhancements, while 2010–2015 indicates two-factor enhancements. The interaction strength significantly increases, rising gradually from 0.16 to 0.45. The joint analysis of these factors provides a better explanation for CLUE. The interaction between CA and AI consistently shows enhancement. In 2000–2005 and 2015–2020, it displays two-factor enhancement, while in 2005–2015, it exhibits nonlinear enhancement. The interaction

value gradually increases from 0.25 to 0.37, resulting in an improved explanatory power for CLUE.

From 2000 to 2005, the interaction between FRAC_AM and PD is 0.17, indicating a nonlinear weakening. From 2005 to 2015, the interaction value increases to 0.19, and it becomes a two-factor enhancement. From 2015 to 2020, the interaction value rises to 0.57, indicating nonlinear enhancement. The joint analysis of these factors provides a more comprehensive explanation of CLUE. From 2000 to 2005, the interaction between FRAC_AM and AI shows single-factor enhancement, and in the remaining periods, it exhibits two-factor enhancement, gradually increasing the explanatory power. Similar to CA and FRAC_AM, PD and AI demonstrate single-factor weakening from 2000 to 2005, with an interaction value of 0.13. However, from 2005 to 2020, the interaction significantly increases, gradually rising from 0.13 to the highest value of 0.59 in the latest period, indicating the strongest interaction value.

## 4. Discussion

Spatial and temporal evolution of CLUE in Henan Province: From 2000 to 2020, the mean value of CLUE fluctuated in 104 counties in Henan Province. From 2000 to 2005, CLUE showed a downward trend, which may be because with the advancement of urbanization, farmers have a strong willingness to go out to work, the agricultural labor force is insufficient, and the enthusiasm for agricultural production is not high. In 2006, after China abolished the agricultural tax, farmers' enthusiasm for farming was enhanced. At the same time, the improvement of agricultural technology promoted the increase of agricultural output and output value. This is the reason why CLUE rose rapidly in 2005–2010. From 2010 to 2015, CLUE declined. This may be because, with the gradual improvement of industrial production levels, the reduction of production costs of pesticides and chemical fertilizers also reduced the opportunity cost of farmers' use, resulting in the gradual or even complete replacement of chemical fertilizers and pesticides. The application of farmyard manure has gradually increased carbon emissions and soil pollution. On the other hand, coupled with the accumulation of "reversed ecological" effects in the production of cultivated land for many years, it has further inhibited the improvement of the CLUE. From 2015 to 2020, the growth trend of CLUE may be a response to the implementation of the Chinese government's "13th Five-Year Plan" for agricultural energy conservation, elimination of old agricultural machinery, and promotion of agricultural energy-saving machinery. Henan Province fully implements high-standard farmland construction to improve land production capacity and encourages the use of organic fertilizers and new agricultural machinery to promote the transformation of agriculture into eco-friendly agriculture. CLUE was highest in southern Henan Province, followed by southwest and northern regions. There are many large grain-producing counties in these areas, and the efficiency of grain production should continue to be improved to reach the leading level in the country. The CLUE values in the central and eastern regions of Henan Province were low, but most of them showed an upward trend in the recent period, especially in the eastern regions. As the most economically developed area in the central part of Henan Province, the planting industry is shrinking and neglected. Additional efforts are needed to improve CLUE and narrow the gap with other areas.

The dynamic relationship between area index and CLUE: Generally speaking, larger arable land means more input and output. CA has a significant positive effect on CLUE as a whole; that is, the larger the total area of cultivated land, the higher the CLUE value. There is significant spatiotemporal heterogeneity in the relationships between cultivated land area and CLUE. However, as time goes by, the area of the negative influence of cultivated land gradually shrinks, but the overall explanatory power of CLUE gradually decreases. From the perspective of input and output, the increase in farm scale is related to the reduction of unit fertilizer and pesticide usage [1], which is beneficial to the improvement of CLUE. In the practice of arable land production, larger land areas are more dependent on the widespread use of machinery, better transportation systems, and advanced agricultural

technology [49], and the increase in CA is important for increasing production while reducing input costs and carbon emissions [35]. Units of agricultural inputs may also be reduced due to the existence of economies of scale. In countries and regions with large plains or relatively flat terrain, such as Henan Province, changes in the patch area of cultivated land will not cause drastic changes in elevation, irrigation, and transportation, which also shows that its positive impact on CLUE is more reliable. However, the large area of cultivated land may promote the deposition of chemical fertilizers and pesticides in the soil, causing soil pollution, and the large-scale use of agricultural machinery may also cause more carbon emissions. However, with the popularization of organic fertilizers, the gradual upgrading of agricultural machinery and agricultural technology, and the enhancement of farmers' awareness of the ecological environment, the area of cultivated land is of great significance for improving CLUE and maintaining national and regional food security.

The dynamic relationship between shape metrics and CLUE: It has been observed that complex shapes of cultivated land tend to inhibit the CLUE. Specifically, when the shape of a cultivated land patch is more regular, the CLUE tends to increase at a faster rate. The regression analysis revealed that shape factors have a greater influence on CLUE compared to area factors. Complex cultivated land patches pose challenges in terms of boundary management and maintenance. On the other hand, regular patches facilitate the development of agricultural machinery services, leading to reduced production costs and labor input. It becomes more difficult and expensive to manage complex patch shapes, as it often requires the construction of road networks, boundaries, or fences [50,51], which ultimately leads to higher production management costs. Flatter cultivated land patches are typically associated with a convenient transportation system, making it easier for growers to manage their operations [52]. Moreover, these patches lend themselves to the efficient use of agricultural machinery in various production activities such as sowing, irrigation, harvesting, and straw returning [53]. Additionally, relatively regular cultivated land patches facilitate land consolidation and are attractive to professional planting contractors. However, the increasing popularity of drone-based applications, such as spraying pesticides and fertilization, and the presence of convenient transportation systems have gradually weakened the inhibitory effect of complex cultivated land patches. The improved CLUE now relies more on advanced industrial equipment and efficient management practices. Nevertheless, considering that complex cultivated land patches are not conducive to improving CLUE, each region needs to prioritize sorting out such patches or simplifying and regularizing them as part of land consolidation efforts.

The dynamic relationship between aggregation metrics and CLUE: In general, the agglomeration of cultivated land contributes to the improvement of CLUE. However, these aggregation indicators exhibit strong spatial heterogeneity. The fragmentation of cultivated land has both positive and negative effects, but over time, the overall impact intensity has weakened. There are several reasons behind this. On the one hand, the fragmentation of cultivated land encourages farmers to refine their cultivation practices, promotes crop diversification, and catalyzes the development of professional farmers and specialized agricultural villages. At the same time, the more concentrated cultivated land may be due to the concentrated use of chemicals such as fertilizers and pesticides, as well as the mechanization of a large amount of fuel, which accelerates soil erosion, causes ecological problems in the cultivated land, reduces agricultural productivity, and increases agricultural carbon emissions. On the other hand, aggregated cultivated land improves mechanical operation efficiency, reduces input costs (such as losses and fuel consumption) per unit area, and facilitates production organization, the large-scale application of irrigation systems, and management, thereby enhancing CLUE [54]. Nevertheless, studies have shown that the positive effect of cultivated land fragmentation cannot be sustained in countries and regions that prioritize agricultural modernization through mechanization and scale, such as Henan Province. In these areas, the negative impact expands gradually. Excessive fragmentation of cultivated land hinders farmers engaging in large-scale and mechanized cultivation, resulting in labor and agricultural inputs being wasted, inconvenient utilization

of agricultural machinery, unclear land ownership, management difficulties, and low efficiency in land use [53]. In response, it is imperative to promote the development of an ecological farmland governance system, which effectively mitigates issues such as land pollution caused by the accumulation of farmland. By doing so, the ecological environment of farmland can be adequately maintained, thereby promoting the overall improvement of CLUE.

In regions where agricultural production is characterized by large-scale mechanization and intensification [55], large and flat cultivated land is advantageous for farmers to carry out cultivation. It helps reduce transportation and labor costs while effectively avoiding the inefficiencies caused by complex land rights [56]. Large and aggregated cultivated land is conducive to large-scale mechanized operations, leading to cost savings for farmers and increased agricultural yield. On the other hand, dispersed and irregularly shaped cultivated land increases transportation and management costs for farmers, intensifies labor and agricultural input consumption, and hinders the improvement of CLUE. Therefore, the interaction between these factors has a stronger explanatory power in the spatial variation of CLUE. Although small and dispersed farmland can increase land income through diversified crop diversity and reduced production risks, the limited economic benefits due to small land areas are not conducive to long-term agricultural development [57]. Consequently, the relationship between land area and farmland patch density gradually shifts towards synergy rather than a trade-off. In essence, the balance between land fragmentation and aggregation represents a trade-off between economic benefits and the ecological environment. For example, specialized villages that plant one or a few types of crops can obtain higher economic returns due to their specialization and scale. However, specially planted crops will often absorb more soil nutrients. Like the four major medicines in Jiaozuo City, after planting once, they cannot be planted again in the next five or even ten years because the land needs time to restore its fertility; secondly, specialized planting reduces biodiversity, leading to a decrease in the ability of farmland ecosystems to resist risks, which will increase the risk to the ecological environment. Therefore, highlighting the significance of seeking a Pareto optimum between agricultural economic benefits and ecological sustainability for the improvement of CLUE.

The study considers both carbon sequestration and carbon emissions perspectives, addressing the shortcomings of previous research [13] that often overlooks the positive effects of land use on the ecological environment. It provides a more comprehensive evaluation of CLUE, offering insights for adjusting agricultural inputs, improving cultivation management practices, and implementing agricultural technology and mechanization in different regions. The findings of CLUE calculations contribute to the balancing of agricultural production processes. For example, reducing inputs can decrease resource consumption and carbon emissions, but reducing inputs may also lead to a decrease in grain yield and carbon sequestration. Previous research has mainly focused on global [58,59] or GWR regression analyses to identify factors influencing CLUE, neglecting the local or temporal non-stationarity, which may lead to inaccurate results. This study takes a spatiotemporal perspective, analyzing the aspects of area, shape, fragmentation, and aggregation from both spatial and temporal angles. It provides a more comprehensive exploration of the spatiotemporal heterogeneity in the driving forces of land use landscape patterns, enabling a more comprehensive analysis and explanation of the dynamic relationship between CLUE and landscape patterns. The study also employs time clustering analysis, identifying four temporal evolution patterns of CLUE which can guide decision-makers in implementing regional management strategies to promote the improvement of CLUE.

## 5. Conclusions

From the perspective of carbon effects, this study utilizes the super-efficiency SBM-GML model to assess the CLUE in 104 counties in Henan Province from 2000 to 2020. Spatial autocorrelation modeling and time series cluster analysis are employed to identify the temporal and spatial characteristics of CLUE. Subsequently, the GTWR model and

interactive detectors are used to analyze the spatiotemporal dynamic relationship between landscape pattern changes and CLUE. The primary conclusions drawn from the analysis are as follows:

(1) The construction of the CLUE evaluation index system should not underestimate or ignore the positive effects of crops on the ecological environment, including carbon sequestration. Comparing the index system that considers both positive and negative crop carbon effects with the results of ignoring carbon effects or positive carbon effects, the study reveals that carbon effects contribute to the improvement of CLUE, and disregarding crop carbon sequestration underestimates the actual carbon effect.

(2) During the study period, the average value of CLUE in Henan Province showed an upward trend, and the value in most years was between 0.50 and 0.70, indicating that the CLUE in most counties in Henan was not high, and there is still great potential for improvement in the future. The overall spatial pattern is weak in the middle and east, high-value areas are concentrated in the southern districts and counties, and the west and north perform well. Spatial autocorrelation shows that CLUE has a significant positive spatial dependence, and this effect also fluctuates and increases with CLUE. Local autocorrelation showed that HH clusters were mainly concentrated in Xinyang City, and LL clusters were mainly concentrated in the central and eastern parts of Henan Province. Based on the change in CLUE, four development modes of CLUE in Henan Province were identified, namely, the wave-by-wave rising type, the fluctuating rising type, the rising and gradually weakening type, and the late-rising and improving districts and counties. This will provide a new perspective for the government to implement zoning management according to local conditions. For example, districts and counties with Wave-rising should continue to promote ecological agriculture, sustainable intensive agriculture, and improve the ecological effect of cultivated land. Districts and counties located in the plains that are Continuous-rising and Fluctuating-rising should take advantage of scale to increase production, develop agricultural processing and logistics industries, and promote economic growth. Late-development districts and counties located in mountainous or hilly areas, such as western Henan and southern Henan, can rely on their endowments to focus on the development of small- and medium-scale high-quality agriculture and increase agricultural income with characteristic or high-end products.

(3) The study used the GTWR model to test that the change of CLUE in Henan Province has significant temporal and spatial non-stationarity, and GTWR can more comprehensively and accurately analyze the relationship between CLUE and landscape pattern changes. The landscape pattern was described from the dimensions of area, shape, and aggregation. The study found that the area of cultivated land has a positive effect on the CLUE, and the spatial heterogeneity and intensity of influence gradually weakened over time. The increase in shape complexity and fragmentation hurts CLUE use as a whole. The spatial heterogeneity of cultivated land shape is weak, the negative impact is extensive and significant, and the degree of negative correlation gradually weakens over time. PD has the strongest explanation for CLUE, and the influence has strong spatial heterogeneity. As time goes by, the positive influence area gradually shrinks, and the negative influence area gradually expands, but the explanatory power and spatial heterogeneity gradually decrease. AI generally has a positive impact, with strong spatial heterogeneity.

(4) The interactive detectors showed that there was a clear synergy between the impact of each landscape indicator on CLUE. The interaction between the shape factor and other factors has a stronger impact on CLUE than the interaction between other factors. The interaction between CA, FRAC_AM, and AI is enhanced, indicating that large, flat, and aggregated cultivated land patches are conducive to the improvement of CLUE, and the interaction between PD and other factors has changed from weakening to strengthening, indicating that for PD, the "double-edged sword" effect gradually tends to be a one-way effect. With the strict implementation of the balance policy of

cultivated land occupation and compensation, the occupation and compensation of cultivated land will continue to exist for a long time. The research findings suggest that when planning for cultivated land translocation and compensation, priority should be given to square-shaped plots that simplify the overall shape. In addition, priority should be given to creating larger and more concentrated compensation plots. During the land consolidation process, emphasis should be placed on the pursuit of more regular, larger, and concentrated cultivated land patches to improve CLUE. This approach is crucial for ensuring food security and protecting cultivated land.

**Author Contributions:** Q.L. conceptualized the idea of the entire manuscript and prepared the initial draft, while J.Q. and D.H. provided input towards the development of ideas, revision, and editorial oversight. M.L. and L.S. were involved in the data analysis. All authors have read and agreed to the published version of the manuscript.

**Funding:** This research was funded by the National Natural Science Foundation of China with grant numbers 42071220.

**Institutional Review Board Statement:** Not applicable.

**Data Availability Statement:** The data utilized in this study were collected from pertinent, publicly available websites.

**Acknowledgments:** We thank the reviewers who provided valuable comments to improve the paper.

**Conflicts of Interest:** The authors declare no conflict of interest.

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
