# Peer review of "Spatiotemporal Evolution of Cultivated Land Use Eco-Efficiency and Its Dynamic Relationship with Landscape Pattern Change from the Perspective of Carbon Effect: A Case Study of Henan, China"

_agriculture, doi:10.3390/agriculture13071350_

Round 1
Reviewer 1 Report
Manuscript ID: agriculture-2458227
Title: Spatiotemporal evolution of cultivated land use eco-efficiency and its dynamic relationship with landscape pattern change from the perspective of carbon effect: A Case Study of Henan, China.
This manuscript has evaluated the cultivated land use eco-efficiency that considers both carbon sequestration and emissions using the SBM model at the county level in Henan Province of China. Moreover, the relationship between the cultivated land use eco-efficiency and changes in landscape patterns has been analyzed using Geographically and Temporally Weighted Regression (GTWR) and Interaction Detectors.
But the manuscript has some problems that need to be revised before publishing, as follow below:
- Why are the counties in Henan Province chosen as a case study in this manuscript?
- In the abstract, the research method is not explained in detail.
- Some keywords are repeated in the title of the manuscript.
- Innovation of the research has not highlighted in the introduction.
- In the introduction, research questions/hypotheses about the objectives are not stated.
- The methodology section should be shortened.
- In the results, the explanations of some figures and tables are short.
- The numbering of the figures is wrong.
- In the discussion section, strategy and suggestions should not be given. Suggestions are moved to the conclusion.
- It's better to remove the numbering in the discussion section.
- In the discussion section, it is necessary to answer the research questions/hypotheses and carry out validation.
- The function and international importance of research should be highlighted in the discussion section.
- The discussion section is weak and it is necessary to discuss reasons of the obtained results in more details.
end.
Reviewer 2 Report
The interesting study, well prepared methodically.
1. Too much data in the conclusions. These should be more synthetic conclusions.
2. Does the single product policy really favor organic production (line 339)? What about biodiversity then?
3. Is intensive farming environmentally friendly? (line 559). This is a somewhat dubious statement and requires clarification.
